# Linear System Challenges of Dynamic Factor Models

Brian D. O. Anderson [1,*], Manfred Deistler [2,3] and Marco Lippi [4]

1 School of Engineering, Australian National University, Acton, Canberra, ACT 2601, Australia
2 Research Unit of Econometrics and System Theory, Technische Universität Wien, Wiednerhauptstrasse 8, 1040 Vienna, Austria
3 Department of Mathematics and Statistics, Wirtschaftsuniversität Wien, Welthandelsplatz 1, 1020 Vienna, Austria
4 Einaudi Institute for Economics and Finance, Via Sallustiana, 62, 00187 Rome, Italy
* Correspondence: brian.anderson@anu.edu.au

**Abstract:** A survey is provided dealing with the formulation of modelling problems for dynamic factor models, and the various algorithm possibilities for solving these modelling problems. Emphasis is placed on understanding requirements for the handling of errors, noting the relevance of the proposed application of the model, be it for example prediction or business cycle determination. Mixed frequency problems are also considered, in which certain entries of an underlying vector process are only available for measurement at a submultiple frequency of the original process. Certain classes of processes are shown to be generically identifiable, and others not to have this property.

**Keywords:** dynamic factor model; high-dimensional time series; linear systems theory

## 1. Introduction

The purpose of this paper[1] is to survey (and provide commentary on) a collection of contributions to do with Dynamic Factor Models developed especially by the authors, principally the first two, over more than a decade, taking the ideas as far as a listing of some problems of current interest. Matters which we are especially concerned to highlight include

1.  The theoretical underpinning for allowing modelling to focus on AR rather than ARMA modelling, and the simplifications this makes possible for the identification of models.
2.  Issues arising in practice from numerical problems in attempting to build models; some of these errors can be mitigated through the use of carefully chosen forms for the models.
3.  The difficulties arising from modelling time series of different but multiply related periodicities, e.g., monthly and quarterly, and existing tools for resolving, at least partly, the difficulties. These build on the ideas developed for single-periodicity modelling.

Feeding in all these matters is the application of ideas of digital signal processing, for which issues of handling numerical problems, and handling data streams of multiple periodicities have been well studied. The relevance of ideas of digital signal processing to dynamic factor modelling has perhaps not been fully recognized.

Dynamic Factor Models were introduced two decades ago, see in particular Forni et al. (2000), Forni and Lippi (2001), Stock and Watson (2002), Bai and Ng (2002), with macroeconometric applications as their main purpose. The basic idea was that a small number of factors, common to all the time-series of a large macroeconomic dataset, could be used both in structural modeling and forecasting of some series of interest like inflation, industrial production, unemployment. For a typical dataset, see the monthly US macroeconomic panel, including 128 series, described in great detail in McCracken and Ng (2016).

We begin by recalling and commenting on a number of ideas in a recent survey we coauthored, Lippi et al. (2022). The survey confined attention to the modelling of multivariate time series with the same sampling frequency for every component. As foreshadowed above, in this paper, besides dealing with such material, we treat modelling where time series data can be available with different periodicities, provided they are integrally related, e.g., monthly and quarterly.

Throughout the paper, we shall confine attention to stationary processes.

We take some time to define what is meant by a dynamic factor model, progressively introducing and commenting on the various assumptions that in aggregate constitute the definition. The comments may be as important as the assumptions. The various assumptions reflect reality to some degree, and they also reflect the requirement to formulate a solvable problem. We also offer commentary on the notion of errors; the comments deal with two issues, first the need to in some way limit the errors involved in determining a process model, given that the model is obtained from sample statistics rather than population data, and second, the need to choose a type of process model that is appropriate to the application: one does not want massive errors in whatever it is desired to compute because of sensitivities intrinsic to the model. An overall conclusion (motivated by the signal processing literature) is that on occasions state-variable models may be preferred to ARMA or AR. Accordingly, some discussion is also offered explaining how state-variable models can be determined without even intermediate use of an ARMA or AR model[2]. Analogously to the result that AR models can often be determined from covariance data using the Yule-Walker equations is an algorithm involving a finite number of rational calculations for obtaining state-variable models of a canonical spectral factor from lagged covariance data. There may well be numerical advantages in using such a state-variable model.

The last part of the paper traces developments applicable to multiple frequency systems, or systems in which data of different periodicities is collected. More precisely, we postulate that there is an underlying stationary zero mean vector time series $(y_t) = [(y_t^f)^\top (y_t^s)^\top]^\top$ which is defined at time $t = 0, 1, 2, \ldots$ and with $y_t^f$ (the subvector of "fast" components) observed at all $t$ while $y_t^s$ (the subvector of "slow" components) is observed at $t = 0, J, 2J, \ldots$ for some integer $J > 1$. The task is to build a canonical spectral factor of the process $(y_t)$ using the measured second-order statistics. The whole set-up can be embedded within a high-dimensional time-series framework of course, but this brief statement encapsulates what is different about the mixed frequency problem.

The conclusions section of the paper sets out a number of issues which have not yet been fully addressed.

## 2. Dynamic Factor Models

Dynamic factor models can roughly be described as those associated with a high dimension, say $N$, vector process $(y_t^N)$ (normally taken with zero mean) defined at $t = 0, 1, 2, \ldots$ and possessing an additive decomposition

$$y_t^N = \chi_t^N + \xi_t^N \tag{1}$$

with $(\chi_t^N)$ termed the process of common components or latent variable process, and $(\xi_t^N)$ termed the process of idiosyncratic components. Typically, $(\chi_t^N)$ is regarded as the true process of interest, with $(\xi_t^N)$ some kind of contamination intruding into the measurement process ('noise' in electrical engineering parlance, which in general is colored rather than white). However one possible alternative interpretation is that (1) displays the decomposition of stock price observations into a part $(\chi_t^N)$ representing the comovements between the variables (for instance representing the "market" effect on N different stock returns) and individual movements $(\xi_t^N)$ (representing effects associated with each specific firm). It is assumed that the one-dimensional processes $(\chi_{it}, t \in \mathbb{Z}), i \in \{1, \ldots, N\}$ are also stationary and strongly dependent across the index $i$, while the processes $(\xi_{it}, t \in \mathbb{Z}), i \in \{1, \ldots, N\}$ are stationary and weakly cross-dependent (the terms strongly dependent and weakly

cross-dependent requiring technical definitions). It is standard to allow for variation in $N$ above some $N_0$, with $y_t^N, \chi_t^N$ and $\xi_t^N$ being nested, for example $y_t^N$ is the subvector of $y_t^{N+1}$ formed from its first $N$ entries. Both $(\chi_t^N)$ and $(\xi_t^N)$ are zero mean and the two vector processes are mutually independent. The spectral densities $\Phi_\chi^N(e^{j\omega}), \omega \in [-\pi, \pi]$ and $\Phi_\xi^N(e^{j\omega}), \omega \in [-\pi, \pi]$ are assumed to exist and $(y_t^N)$ is then necessarily stationary with

$$\Phi_y^N(e^{j\omega}) = \Phi_\chi^N(e^{j\omega}) + \Phi_\xi^N(e^{j\omega}). \tag{2}$$

These spectral matrices obviously inherit the nesting property of the underlying processes.

We sum up the above description as follows.

**Assumption 1.** *A nested (with N) set of observation processes $(y_t^N)$ of zero mean arises as an additive combination of a stationary zero mean nested common component process $(\chi_t^N)$ and a stationary zero mean nested process of idiosyncratic components $(\xi_t^N)$, as in (1). The latter two processes are independent, the first is strongly cross-dependent (a term defined subsequently) and the latter is weakly cross-dependent (also defined subsequently), and the spectra of the three processes are related by (2).*

Note that there is no assumption to this point that the spectra are rational, or otherwise finitely parametrized.

To formulate a solvable identification problem, a number of other assumptions are always invoked in considering dynamic factor models, and we will introduce them gradually and attempting a logical progression over the subsequent subsections, albeit with an attempt to justify their relevance.

### 2.1. The Goal of Modelling

Exactly what should be the goal of modelling in science is as much a philosophical as it is a scientific question, with answers often reflecting on the real-world utility of any model. Much the same question arises in connection with economic modelling; the very highly-cited paper Sims (1980) seeks to peer behind the mathematical formalism to understand exactly what relation the models of economics have to the real world.

For the purposes of this paper, we will simply assert that a model may be desired for various purposes which may not all be applicable for any one model at a particular time. Two of these for example might be forecasting of future values of a time series, and identification of business cycles. For modelling and especially forecasting purposes, often a canonical (stable, miniphase) spectral factor of $\Phi_\chi^N(e^{j\omega})$ is desired. Such a spectral factor is to be constructed using the data $y_t^N$ with $0 \leq t \leq T$ where $T$ is some finite positive integer. The dimension $N$ may in fact exceed $T$, indeed theoretical studies focus on letting $N$ rather than $T$ tend to infinity.

To make progress, further assumptions beyond the nesting property are needed. Some of these assumptions bear on the ability to somehow dispose of the contaminating influence of $(\xi_t^N)$, and we will deal with them further below, and at that time clarify the strong and weak dependence notions. So assume temporarily that somehow we can work just with $(\chi_t^N)$.

### The Issue of Errors and Continuity

Observe that there are different ways to describe a stable miniphase spectral factor, see e.g., Hannan and Deistler (2012) and Caines (2018) for the relevant linear system theory background. If it has a rational transfer matrix, state-variable descriptions could be used (and then there is freedom to choose a coordinate basis). There is interest especially in those which are of minimal dimension. Again, if the spectral factor is rational, a left or right matrix fraction description could be used (a left matrix fraction description being an ARMA description, subject to rules about leading coefficient matrices). There is interest especially

in those where the fraction is coprime (common factors are generally as unhelpful in a matrix transfer function as they are for the numerator and denominator of a scalar transfer function). If it is known to be autoregressive, an AR description may be used. A very different form again is provided by a series of graphical images of the values of the transfer matrix entries as a function of $\omega$; this might even be the most convenient form of model if the model does not have a rational transfer function matrix.

No matter what means are used to construct a model for the process $(\chi_t^N)$, it is clear that one needs to be certain that a small error in (say) the lagged covariance sequence values (due say to use of sample rather than population statistics) will ultimately flow through to a small error in the quantities obtained in whatever particular application is envisaged when using the model. Since the model is the tool underpinning the application, it is reasonable to seek further assurance that the small error in the lagged covariance values (which in almost all cases are the raw data from which any model is constructed) will also give rise to a small error in the actual model of the minimum phase transfer function, be it frequency domain description, AR, ARMA, or general state-variable. Thus, if the model is being used for prediction, one ultimately wants a small error in the estimation variance of the predictions, and an accurate AR model may consequently be sought (assuming existence of an AR model is guaranteed). On the other hand, and even if an AR model should exist, if the model is being used to look for spectral peaks capturing some oscillatory phenomenon, or to understand the different bandwidths of the processes of interest, one is more likely to want a small error in the frequency domain description of the model, or even the pole positions of the model, as opposed to the AR coefficients.

Evidently then, some kind of a continuity is being sought (small errors in data used for generating the model should give small errors in the ultimate answer in the application for which the model is being used), but evidently also the particular parameters relevant to considering continuity, including continuity within the model description, are application dependent.

The most common sort of continuity in a model to which attention is given is the continuity of coefficients in a finite dimensionally parameterised model. One thinks of the class of systems of interest as collection of subsets, where the subsets are obtained by performing (data driven) model selection or by making various assumptions about integer parameters associated with the dimension of the innovations process, or a minimal state variable realization. However, such thinking may exclude certain systems at the start, and on occasions continuity of a frequency domain plot may be of more interest than continuity of a parameter in a model of some assumed order.

Because so much work in econometrics is linked to forecasting, there is a justified preference for using (when they exist) autoregressive models. The crucial part of the modelling to get right is to have accurate AR coefficients, which (in the event that Yule-Walker equations are used to derive the coefficients) probably requires a well-conditioned and thus more easily invertible matrix of lagged covariance coefficients, in addition to some suitable level of accuracy in the lagged covariance coefficients themselves. [More precisely, since on occasions a singular matrix may appear in the Yule-Walker equations, one will want a modified form of well-conditioning as indicated below.] One should note that the conditioning of the matrix in question can be related to an important property of the associated multivariate spectrum. Let $\nu_{max}$ denote the maximum over $\omega$ of the maximum eigenvalue of the Hermitian matrix $\Phi_\chi^N(e^{j\omega})$, and let $\nu_{min}$ denote the minimum over $\omega$ of the smallest nonzero eigenvalue of $\Phi_\chi^N(e^{j\omega})$. (Zero eigenvalues arise if the matrix is singular and these are for this purpose discarded; it is assumed the number of zero eigenvalues is the same for all $\omega$). The condition number of the block Toeplitz matrix inverted in a Yule-Walker calculation is known to approach $\nu_{max}/\nu_{min}$ as the size of the matrix goes to infinity, Miranda and Tilli (2000); Serra (1999), with modification in the singular case. Thus, ill-conditioning in a Yule-Walker calculation may be an automatic consequence of the nature of the spectrum being modelled.

In a sense, if we use an information criterion such as AIC for estimating the AR order $p$, such a problem is unlikely to occur Shibata (1980).

In contrast however to the use of a model for prediction, if bandwidth determination and oscillatory phenomena or accuracy of pole positions in the model are issues of concern, it is a well-established, indeed now old, fact in digital signal processing that autoregressive (and ARMA) models (known often in the relevant literature as direct models) can be very dangerous, in the sense that pole positions and frequency peaks can be extremely sensitive to the AR coefficients, e.g., Mitra (2011); Rabiner and Gold (1975). MATLAB (Signal Processing Toolbox) even allows the determination of "lattice-form" state-variable realizations of a prescribed transfer function that will have low or even minimum sensitivity in the frequency response to truncation or round-off errors associated with the coefficients in the realization, or minimum round-off errors in calculations using the coefficients in the realization to compute outputs from inputs.

Using an AR representation for the canonical spectral factor will likely be a bad idea in this situation: the numerical parameters in the model may need to be known to an unreasonable degree of accuracy. At the same time, as the signal processing literature indicates, there may be a state-variable model for which much less demanding accuracy requirements apply to the numerical parameters in the model, in order to learn from the model what is desired.

Note further that if multi-step prediction is in contemplation, rather than one-step-ahead prediction, recursion using AR coefficients will also be risky for those systems with pole positions which are very sensitive functions of the AR coefficients. In the event though that prediction was required for a certain fixed number of steps, $h$ say, ahead, as opposed to an interval extending some distance into the future, one could potentially avoid recursion by directly estimating an $h$-step prediction model by minimizing the $h$-step prediction errors by least squares. See for example Bhansali and Ghosh (1999), which distinguishes the two approaches with the names "plug-in method" and "direct method".

To sum up then, we must be concerned about two things:

1.  If there is a true underlying process with an associated canonical spectral factor, and a canonical spectral factor obtained using approximate values of lagged covariance coefficients, there will be an error between the two.
2.  We should use a description of the computed canonical spectral factor that is appropriate to the application, in the sense that the error mechanisms in the relevant computations will not cause significant errors in the quantities computed as part of the application.

In connection with the first issue of continuity of the canonical spectral factor frequency response, reference Anderson (1985) provides a sufficient condition for continuity: the derivative (with respect to $\omega$) of $\Phi_\chi^N(e^{j\omega})$ must be bounded, to ensure that a small $L^\infty$ error in the spectrum gives rise to a small $L^\infty$ error in the canonical spectral factor frequency response. To relate this to a covariance sequence condition, observe that

$$\Phi_\chi^N(e^{j\omega}) = \sum_{s=-\infty}^{\infty} \mathbb{E}[\chi_{t+s}^N (\chi_t^N)^\top] e^{-j\omega s}.$$

Formal differentiation yields

$$\frac{d}{d\omega} \Phi_\chi^N(e^{j\omega}) = \sum_{s=-\infty}^{\infty} (-j)s \mathbb{E}[\chi_{t+s}^N (\chi_t^N)^\top] e^{-j\omega s}$$

and by the Weierstrass M-test, uniform convergence in $\omega$ will occur provided

$$\sum_{s=-\infty}^{\infty} s \|\mathbb{E}[\chi_{t+s}^N (\chi_t^N)^\top]\| < \infty.$$

A sufficient condition ensuring this property is incorporated in the following:

**Assumption 2.** *The derivative (with respect to $\omega$) of $\Phi_\chi^N(e^{j\omega})$ is bounded, which is guaranteed if the covariance sequence $\mathbb{E}[\chi_{t+s}^N(\chi_t^N)^\top]$ goes to zero faster than $(1/s)^\alpha$ for any $\alpha > 2$.*

As explained in Anderson (1985), in the absence of a derivative bound, an arbitrarily small variation of a scalar spectrum can produce an arbitrary phase shift anywhere in $(0, \pi)$ in the associated spectral factor (though the spectral factor magnitude is obviously simply given by taking the square root of the spectrum value at each frequency, and so continuity applies to the magnitude). Evidently, without the Assumption, the spectral factorization problem is almost ill-posed for any real world situation.

As noted above, this assumption is relevant to securing accuracy in frequency domain behavior of the miniphase spectral factor. It is at best indirectly relevant to securing accuracy in say an AR model being used for prediction purposes, if such is known to exist. Nevertheless, if by virtue of some other assumptions or argument, a *stable* AR model (and surely stability is a requirement) exists, then this imposes an exponential rate of decay (with increasing lag) on the lagged covariances, and the assumption is fulfilled. Thus, the assumption is not offensive to custom. Actually, the ratio $\nu_{max}/\nu_{min}$ is also partially relevant to the assumption: the greater is this ratio, the more sensitive appears to be the spectral factor frequency response to variations in the power spectrum, Anderson (1985).

### 2.2. Controlling the Number of Parameters to Be Estimated

It is clear that any canonical spectral factor of practical use has to be parametrised by a finite number of quantities, be they integers or real parameter values. One way to ensure this can occur is simply to lay down an assumption that the spectrum is rational:

**Assumption 3.** *For each N, $\Phi_\chi^N(e^{j\omega})$ is rational and therefore bounded in $e^{j\omega}$.*

The boundedness assumption rules out the inclusion of a deterministic component.

On the face of it, a rationality assumption is a leap of faith–there seems no reason grounded in physics or economics that suggests this is reasonable, apart perhaps from a belief in the general applicability of an Occam's Razor principle. What makes it reasonable and less instrumentalist though is to recognize that a very minor and more reasonable variant on it will be satisfactory. Using the crucial fact that any covariance sequence satisfying Assumption 2 (corresponding to a stationary process) can be approximated arbitrarily closely in $L^p$ norm for any $p \in [1, \infty)$ by a covariance with rational spectrum[3], it follows that if the true spectrum is not rational, under the decay condition of Assumption 2, the rationality assumption of Assumption 3 becomes in practical terms acceptable, as it can apply to a spectrum arbitrarily close to that of the true process. There is however a caution: while the approximation idea is valid for a fixed $N$, the assumption says nothing about what happens when $N$ goes to infinity, and the next and less innocent assumption given below is needed to address the issue.

More specifically and given Assumption 3, there will in fact be three critical $N$-dependent integers associated with the spectral factor of $\Phi_\chi^N(e^{j\omega})$:

1. The dimension $n$ of a minimal state-space realization of the canonical spectral factor (or its McMillan degree)
2. The number of columns $q$ in the minimal state-space factor, or the dimension of an innovations sequence (this being a white noise sequence, known in this context as the dynamic factor sequence)
3. The rank $r$ of $\gamma^N(0) := \mathbb{E}[\chi_t^N(\chi_t^N)^\top]$, which is the dimension of the space spanned by $\chi_{1t}, \chi_{2t}, \cdots, \chi_{Nt}$.

While the existence of $n$ (for fixed $N$) is a direct consequence of Assumption 3, the other two integers deserve some comment. The rationality assumption implies there is a rational canonical spectral factor for $(\chi_t^N)$, thus

$$x_{t+1} = Fx_t + Gw_{t+1} \tag{3}$$
$$\chi_t^N = H^N x_t,$$

where $F \in \mathbb{R}^{n \times n}, G \in \mathbb{R}^{n \times q}$ with rank $q$, $H^N \in \mathbb{R}^{N \times n}$ with rank $r$, and the matrices $H^N$ are nested; this means we can write for $n$-dimensional row vectors $h_1, h_2, \ldots,$

$$H^N = \begin{bmatrix} h_1 \\ h_2 \\ \vdots \\ h_N \end{bmatrix} \tag{4}$$

The process $(w_t)$ is zero mean white noise with $\mathbb{E}[w_t w_t^\top] = I_q$. Stability and the miniphase property correspond to

$$\rho(F) < 1, \quad \text{rank} \begin{bmatrix} I - Fz & -G \\ H^N & 0 \end{bmatrix} = n + q \text{ for } |z| \leq 1 \tag{5}$$

We note that one can also write

$$\chi_t^N = H^N (I - Fz)^{-1} Gw_t = K^N(z) w_t \tag{6}$$

with obvious definition of the $N \times q$ transfer matrix $K^N(z)$. As is well known, such a stable and miniphase transfer function satisfying the spectral factorization equation $\Phi_\chi^N(e^{j\omega}) = K^N(e^{j\omega})(K^N(e^{-j\omega}))^\top$ is unique up to right multiplication by a real orthogonal matrix, here of dimension $q \times q$, but through normalization can be made unique.

Since $GG^\top$ is of size $n \times n$, and the number of columns of $G$, viz. $q$, is minimal in a canonical spectral factor, it is immediate that $q \leq n$. Further, since

$$\gamma^N(0) = \mathbb{E}[H^N x(0) x^\top(0)(H^N)^\top],$$

we see that $r$ is bounded by the dimension of the square matrix $\mathbb{E}[x(0)x^\top(0)]$, i.e., $r \leq n$. However, also, $\gamma^N(0)$ is related to the spectrum by

$$\gamma^N(0) = \frac{1}{2\pi} \int_{-\pi}^{\pi} \Phi_\chi^N(e^{j\omega}) d\omega$$

which shows that $r \geq q$.

All three integers evidently depend on $N$, and one of our interests is to let $N$ vary. However, using these integers, we introduce by fiat an almost natural strengthening of the nesting property of the underlying processes and their spectral densities and thereby almost 'define away' the dependence on $N$:

**Assumption 4.** *There exists a positive integer $N_0$ such that for all $N \geq N_0$, the integers $n, q, r$ are independent of $N$.*

By way of heuristic explanation or even justification, we comment that as $N$ increases, the parameter count in the canonical spectral factor increases, linearly with $N$. However, the assumption means that the *number of independent driving signals which are the ultimate cause of the output of the canonical spectral factor (the dimension of what is termed the dynamic factors of the model) remains constant* at $q$, the *complexity* (minimal state-space dimension) $n$ of the model stays constant, and once the output vector has become sufficiently big (of size $r$), *all further outputs are linear combinations of other ones*, and not even delayed versions

of others. There is almost no new information in these further outputs[4], other than that explaining how they are related to earlier outputs. Implicit then in the assumption is a sort of Occam's Razor principle, limiting the rate of increase in complexity (the parameter count effectively) of the model to the least amount consistent with increase in $N$.

A physical analogy arises if one considers $\chi_t$ as like a vector of vibrations arising in a piece of machinery, or electrical thermal noise at the ports of a linear circuit. There may be a limited number $q$ of sources of the vibrations (resistors in the circuit analogy), and one could add more and more sensors (corresponding to $N$) to the set-up. Their presence would not change the underlying dynamical behavior of the machinery or circuit, governed by a set of differential equations with 'complexity' $n$. At some point when adding sensors (but with no associated sensor noise since that is to be bundled in with $\xi_t$ in this analogy), say when $N = r$, essentially no new information is provided by an additional sensor: the additional sensor just provides a linear combination of the measurements from other sensors.

The assumption that $q$ is independent of $N$ is shared by all works on dynamic factor models. A large majority in that literature also shares the assumption that $r$ (and $n$) is independent of $N$. However, as the following simple example shows, $q$ can be independent of $N$ without $r$ being independent of $N$. Let

$$\chi_{it} = (1 - \alpha_i z)^{-1} w_t = w_t + \alpha_i w_{t-1} + \cdots,$$

where $w_t$ is a scalar white noise, so that in this case $q = 1$. Now, if the $\alpha_i \in [0, .9]$ and $\alpha_i \neq \alpha_j$ for all $i \neq j$, then the dimension of the space spanned by $\chi_{it}, i = 1, \ldots, N$ is $N$, so that Assumption 4 does not hold, see Forni et al. (2015) for details.

Henceforth, we shall neglect the dependence of $n, q, r$ on $N$ for small values of $N$.

We summarize the inequalities linking the three key integer parameters and the minimum allowed value $N_0$ of $N$ in the following theorem:

**Theorem 1.** *With notation as above and under Assumptions 2–4, there holds $q \leq r \leq n$ and $r \leq N_0$, (with $r, q, n$ independent of $N \geq N_0$).*

Actually, $r > q$ has always been observed in empirical applications to large macroeconomic data sets, Barigozzi et al. (2021). One can observe also that as $N$ grows, the number of parameters grows linearly with $N$, in fact as $O(nN)$.

Given covariance data for $(\chi_t^N)$, one way the integer $r$ can be obtained is to use the fact that for large enough $N$,

$$r = \text{rank } \mathbb{E}[\chi_t^N (\chi_t^N)^\top]$$

Sample averages have to replace the population averages, and methods such as singular value decomposition could be used to execute the calculations.

For future reference, we note that the equation

$$\Phi_\chi^N(e^{j\omega}) = K^N(e^{j\omega})(K^N(e^{-j\omega}))^\top$$

implies that the McMillan degree of $K^N$ (the dimension of a minimal realization of $K^N$, i.e., $n$) is one half the McMillan degree of $\Phi_\chi^N(e^{j\omega})$, or equal to the McMillan degree of the transfer matrix with associated impulse response the lagged covariance sequence $\gamma^N(0), \gamma^N(1), \ldots$, with $\gamma^N(s) = \mathbb{E}[\chi_t^N (\chi_{t-s}^N)^\top]$, see, e.g., Anderson (1969).

*2.3. Strong and Weak Cross-Dependence*

Strong dependence captures the notion that as $N$ tends to infinity, the effect of the input signals, and all the dynamic complexity showing up in the state variable, continue to show up in a nondecreasing way in the progressively added output components. This can be captured formally by

**Assumption 5.** *For all* $\omega \in [-\pi, \pi]$, *the largest* $q$ *eigenvalues of* $\Phi_\chi^N(e^{j\omega})$ *diverge linearly with* $N$ *as* $N \to \infty$, *while the other eigenvalues are zero. In addition, the* $r$ *largest eigenvalues of* $T^{-1} \sum_1^T \chi_t^N (\chi_t^N)^\top$ *diverge to infinity linearly with* $N$, *while the other eigenvalues are bounded.*

Note that the second part of the assumption is not a consequence of the first part, Lippi et al. (2022).

Weak dependence in a sense is the converse. The different components of $\xi_t^N$ do not 'reinforce' one another due to some common dependences, but rather have very limited dependence between each other:

**Assumption 6.** *The largest eigenvalue of* $\Phi_\xi^N(e^{j\omega})$ *is bounded for all* $N$ *and all* $\omega \in [-\pi, \pi]$.

Evidently, for any fixed $T$, the larger $N$ becomes, the greater is the 'signal-to-noise' ratio between $\chi_t^N, t \in \{0, 1, 2, \ldots, T\}$ and $\xi_t^N, t \in \{0, 1, 2, \ldots, T\}$. Motivated by communication engineering, $\lambda_{max}(\Phi_\chi(e^{j\omega}))/\lambda_{max}(\Phi_\xi(e^{j\omega}))$ can be regarded as a surrogate for signal-to-noise ratio. This might prompt us to contemplate the possibility of 'denoising', i.e., separating out the $(\chi_t^N)$ process from $(y_t^N)$ for large enough $N$. Note that in using terminology with the word 'noise', we acknowledge that this interpretation of the $(\chi_t^N)$ process will not be appropriate in some applications.

There are similar but not identical ways to introduce assumptions of this type which, taken in conjunction with the earlier assumptions, are just as suitable. Discussion with comparison can be found in Lippi et al. (2022).

This concludes the setting out of assumptions on the processes being examined. We turn now to the procedures used for modelling.

### 3. Obtaining the Common Component Process from the Measurement Process

The key to achieving an additive decomposition of the measurement process is to appeal to the strong dependence of the processes $(\chi_{it}^N)$ and the weak cross-dependence of the processes $(\xi_{it}^N)$.

It was shown in Stock and Watson (2002), Forni et al. (2009) that under the earlier assumptions a consistent estimator of $\chi_t^N$ can be obtained by using a static PCA of $y_t^N$. At a similar time, Bai and Ng (2002) demonstrated consistent estimation of $r$; the work relies also on using asymptotic principal components, is nonparametric, and does not require a low value of $n$. As for the input dimension of the canonical spectral factor of the common component process, Hallin and Liška (2007), Amengual and Watson (2007), Bai and Ng (2007), Onatski (2009) allow estimation of $q$.

As noted in Forni et al. (2009), it is evident that for all $N \geq N_0$, the dimension of the space, call it $\mathcal{S}_t^N$, spanned by $\chi_{it}, i = 1, 2, \ldots, N$ is $r$. Let $(f_t)$ be an $r$-dimensional process such that $f_t$ forms a basis for $\mathcal{S}_t^N$ and $f_t = S\chi_t^N$ where $S$ is an $r \times N$ selector matrix independent of $t$ and picking out a spanning set of entries[5] of $\chi_{it}$ for $i$ limited to the set $i = 1, 2, \ldots, N_0$. It follows also that

$$\chi_t^N = L^N f_t$$

for some factor loading matrix $N \times r$ matrix $L^N$ independent of $t$. The entries of $f_t$ are often termed *minimal static factors*.

Actually, in place of the selector matrix $S$, one can use any left multiple $TS$ with $T$ nonsingular and $r \times r$.[6] A minimal static factor is then seen to be unique up to premultiplication by a constant nonsingular matrix $T$, with the factor loading matrix unique up to postmultiplication by $T^{-1}$. Note that the way $L^N$ has been constructed, the successive (with $N$) matrices are nested. It is possible to eliminate the nonuniqueness using normalizing procedures, see, e.g., Lippi et al. (2022).

The connection with the earlier model of (3) is straightforward to see. Evidently, for some $r \times n$ matrix $\tilde{H}$, there holds

$$f_t = \tilde{H} x_t \tag{7}$$

with

$$H^N = L^N \tilde{H} \tag{8}$$

The process $(f_t)$ in a sense is now the real process of interest. Of course, its dimension does not grow with $N$. Its determination however depends on using large $N$, in order that $(\chi_t^N)$ can be extracted from $(y_t^N)$, and then $(f_t)$ is extracted from $(\chi_t^N)$.

Importantly, it now becomes computationally practical to consider the determination of the integer $n$ from the lagged covariance sequence determined by $f_t$. By way of background, recall, see, e.g., Hannan and Deistler (2012); Ho and Kalman (1966) that $n$ is available as the rank of all sufficiently large square block Hankel matrices formed from the covariance data

$$\mathbb{E}[f_t f_t^\top], \mathbb{E}[f_t f_{t-1}^\top], \mathbb{E}[f_t f_{t-2}^\top], \ldots$$

Singular value decomposition (SVD) is a computationally effective way to determine the rank. Of course, given that the covariances are actually determined using sample rather than population averages, and using finite rather than infinite values of $N$ and $T$ to extract $(\chi_t^N)$ from $(y_t^N)$, judgment will have to be used to deal with the computational errors; SVD really is a tool for estimating rather than determining the rank. Further details appear in a later section.

*Two Alternative Approaches to Additive Decomposition of the Spectral Matrix*

For completeness, we summarize here two ways, see Lippi et al. (2022), whereby the spectra $\Phi_\chi^N(e^{j\omega})$ and $\Phi_\xi^N(e^{j\omega})$ can be derived from $\Phi_y^N(e^{j\omega})$.

First, an assumption is made that the idiosyncratic process vector is a vector of independent second order processes. Under such a special assumption, it is in principle possible using partial fraction expansion of the power spectrum $\Phi_y^N(e^{j\omega})$ to separate the two spectra $\Phi_\chi^N(e^{j\omega})$ and $\Phi_\xi^N(e^{j\omega})$. Second, dropping the rationality assumption but retaining the assumption that the idiosyncratic process vector is a vector of independent second order AR processes, and by considering $(q+1) \times (q+1)$ submatrices of $\Phi_\chi^N(e^{j\omega})$ which contain precisely one diagonal entry (this requires $N > 2q$), and which therefore have all entries known from $\Phi_y^N(e^{j\omega})$ except for that diagonal entry, the singularity of the matrix generically enables recovery of the diagonal entry.

The diagonality assumption on $\Phi_\chi^N(e^{j\omega})$ in both procedures is stronger than the weak cross-dependence and can be seen as a misspecification of the general model. However, Doz et al. (2012) argues that: (i) under diagonal $\Phi_\xi^N(e^{j\omega})$ the model can be estimated by Maximum Likelihood, (ii) as $N$ tends to infinity the effect of the misspecification vanishes. Whether the effect also vanishes in the procedures outlined above is unknown.

## 4. Tall Stable Miniphase Spectral Factors and Singular Rational Spectra

An important property is (for $N > q$) the tallness of the transfer function matrix $H^N(I - Fz)^{-1}G$, which is linked with the singularity of $\Phi_\chi^N(e^{j\omega})$. Likewise, the transfer function matrix between the innovations and the minimal static factor process $(f_t)$, being given by $\tilde{H}(I - Fz)^{-1}G$ is $r \times q$, and while always $r \geq q$, as noted earlier $r > q$ is normally encountered in practice.

In this section, we recall results appearing in Anderson and Deistler (2008, 2009); Anderson et al. (2012b); Chen et al. (2012) stemming from this property. The first key result deals with the zeros of the generating model.

**Theorem 2** (Anderson and Deistler (2008))**.** *Consider the stable, miniphase model* (3) *with* $N > q$ *and assume that the entries of the matrices* $F, G, H^N$ *take generic values. Then the model has the zero free property*[7]

$$rank \begin{bmatrix} I - Fz & -G \\ -H^N & 0 \end{bmatrix} = n + q \; \forall \, z \tag{9}$$

In Theorem 2 we make the assumption that the entries of $F, G, H^N$ take generic values, with the requirements of stability and the miniphase property. The inclusion of these requirements is dealt with further below. Genericity of course can only be defined with respect to a well specified parameter space. In Theorem 2, in agreement with the Anderson and Deistler's papers mentioned above, the parameter space is implicitly defined as a minimum-restriction set. Precisely, the parameter space is an open subset of $\mathbb{R}^g$, contained in $\mathbb{R}^g - G$, where $g$ is the total number of entries of $F, G, H^N$ and $G$ is the subset of $\mathbb{R}^g$ where either the stability or the miniphase condition does not hold. If the system is zero free, that is consistent with the miniphase requirement, and it is clear that if the system is zero free for one set of parameter values, it will be free for almost all, given that it is a rank condition. (Note also that an inessential variant to the count can occur if some of the entries are fixed at zero, or one, as can occur if a square matrix is a companion matrix or a block companion matrix for example. Such fixed entries are not part of the count.) In other words, apart from the stability and miniphase conditions, we assume that each parameter can vary independently of the others. Throughout the paper genericity will always be implicitly defined with respect to such minimum-restriction parameter spaces. See however the Conclusions for further considerations on this issue.

Zero freeness with $N > q$ implies a number of properties:

1.  Suppose that for polynomial matrices $\bar{A}^N(z) \in \mathbb{R}^{N \times N}[z]$, $\bar{B}^N(z) \in \mathbb{R}^{N \times q}[z]$ with $\bar{A}^N$ and $\bar{B}^N$ left coprime[8] there holds $K^N(z) = (\bar{A}^N)^{-1} \bar{B}^N$. Then zero freeness implies $\bar{B}^N \in \mathbb{R}^{N \times q}$ has full rank for all $z$. Accordingly, there exists a polynomial unimodular (constant determinant) common premultiplier $U^N(z)$ of $\bar{A}^N, \bar{B}^N$ such that a new polynomial fraction description of $K^N(z)$ exists with

    $$K^N(z) = (U^N(z)\bar{A}^N(z))^{-1}(U^N(z)\bar{B}^N(z)) = (A^N(z))^{-1}B^N$$

    where $B^N$ is a *constant* matrix. We have stressed earlier that $N$ can increase; note though that $\det A^N(z)$ will have degree $n$ as a polynomial in $z$ for all $N \geq N_0$.

2.  The system (3) is left invertible with unknown initial state, that is, there exists an integer $L \leq n$ such that from the sequence $\chi_k^N, \chi_{k+1}^N, \dots, \chi_{k+L-1}^M$ the state $x_k$ and the sequence $w_{k+1}, w_{k+2}, \dots, w_{k+L-1}$ can be determined.

Other characterizations and properties to be found in Anderson and Deistler (2008) deal with the Smith-McMillan form, the impulse response of $K^N(z)$, and the absence of nontrivial so-called output-nulling subspaces.

There is of course a very minor restatement of the above Theorem and the two following comments based on the canonical spectral factor generating $f_t$ rather than $\chi_t^N$. It too is (normally) zero free, due to the tallness inequality $r > q$. However, in working with an AR description of $f_t$ and setting $\chi_t^N = L^N f_t$, rather than working with an AR description of $\chi_t^N$, the reduction in parameter count is substantial, and varies as $O(N)$ rather than $O(N^2)$.

To this point, the stability of the model (3) and its connection with a spectrum have not been appealed to. To bring in those ideas, continue to assume the underlying stable miniphase spectral model of $\Phi_\chi^N$ is (3) and the matrices in a state-variable description are generic.

**Theorem 3** (Anderson and Deistler (2008, 2009); Anderson et al. (2012b); Chen et al. (2012)). *Adopt the same hypotheses as Theorem 2. Then*

1.  *The spectrum $\Phi_\chi^N$ can be generated by a singular autoregressive process*

    $$[I + A_1^N z + A_2^N z^2 + \cdots + A_m^N z^m]\chi_t^N = B^N w_t \tag{10}$$

    *where $B^N \in \mathbb{R}^{N \times q}$ has rank $q$ and $w_t$ is a unit covariance innovations process; the $A_i^N$ and $B^N$ are computable in a finite number of rational calculations from the lagged covariances $\mathbb{E}[\chi_t^N (\chi_t^N)^\top], \mathbb{E}[\chi_t^N (\chi_{t-1}^N)^\top], \dots$.*

2. Suppose the rational spectral matrix $\Phi_\chi^N(e^{j\omega})$ is written in the form

$$\bar{H}^N(I - z\bar{F})^{-1}\bar{Q}^N + (\bar{Q}^N)^\top(I - z^{-1}\bar{F}^\top)^{-1}(\bar{H}^N)^\top - \bar{H}^N\bar{Q}^N, \quad z = e^{j\omega}$$

where the triple $\{\bar{F}, \bar{H}^N, \bar{Q}^N\}$ is minimal (controllable and observable) in the state-variable sense and all eigenvalues of $\bar{F}$ have magnitude less than 1, such matrices being computable in a finite number of rational calculations from the lagged covariances $\mathbb{E}[\chi_t^N(\chi_t^N)^\top]$, $\mathbb{E}[\chi_t^N(\chi_{t-1}^N)^\top]$, .... Then

   (a)   there exists $\bar{G}^N$ such that the stable miniphase (canonical) spectral factor can be written in the form $\bar{H}^N(I - z\bar{F})^{-1}\bar{G}^N$

   (b)   $\bar{G}^N$ is computable in a finite number of rational calculations from $\bar{F}$, $\bar{H}^N$ and $\bar{Q}^N$, and

   (c)   the triple $\{\bar{F}, \bar{H}^N, \bar{G}^N\}$ is minimal.

3. In the event that $\bar{G}^N$ defined above is independent of N, say $G = \bar{G}^N$, then $\bar{Q}^N = P(\bar{H}^N)^\top$ for some positive definite P which is independent of N, with P the unique solution of $P - \bar{F}P\bar{F}^\top = GG^\top$.

The following statement is a quite obvious consequence of the previous Theorems:

**Theorem 4.** *Consider the ARMA process*

$$A(z)x_t = B(z)u_t,$$

*where $A(z)$ is an $m \times m$ stable polynomial matrix, $B(z)$ is an $m \times q$ polynomial matrix. Suppose that $m > q$, so that $x_t$ is a singular ARMA process. Then for generic values of the coefficients of the entries of $B(z)$, $B(z)$ is zeroless, so that $x_t$ has a finite AR representation.*

The relevance of AR processes in our context is not a new idea of course, see Stock and Watson (2005), Forni et al. (2009), Stock and Watson (2016), Forni et al. (2020). These papers apply Theorem 4, in which $x_t$ is replaced by $f_t$, while singularity is a consequence of the assumption that $r > q$, so that we are again working with a generically zero-free model.

The first statement of the Theorem 3 above is little more than a restatement of the first statement of Theorem 2. As a side comment, note that $B^N$ can be uniquely specified by multiplying on the right by an orthogonal matrix rendering it triangular. Reference Anderson et al. (2012b) contains further information concerning exploitation of the freedoms that remain after such a normalization, and the associated parameter counts, including the value of $m$. The row degrees in particular of the matrix $I + A_1^N z + \cdots + A_m^N z^m$ can be minimized in a certain sense. Notwithstanding, the number of nonzero parameters in the singular AR description is essential proportional to $N^2$, which is an unattractive feature, contrasting with the linear dependence on $N$ of the parameter count for a state-variable model. In a different direction, a different aspect of the awkwardness of working with the singular AR model is that the nesting property for $N \geq N_0$ is not simply captured. Of course, in the event one seeks, as noted above, an AR model for $(f_t)$ rather than $(\chi_t^N)$ the difficulty of working with large $N$ is no longer present. Not only does one have $\chi_t^N = L^N f_t$, with $N$ is now responsible for only linear growth in the number of parameters, but the nesting property holds, in particular being handled by $L^N$.

The second statement is more subtle: the fact that the same pair $\bar{F}$, $\bar{H}^N$ can be used for state variable realizations of the covariance sequence associated with $\Phi_\chi^N(e^{j\omega})$ and the associated stable miniphase spectral factor is an old though not well-known result, see, e.g., Anderson (1969); it is not restricted to the zero-free (autoregressive) case. The multivariate version of the result is not easy to prove, in contrast to the converse (which states that a triple 'realizing' the covariance sequence can inherit two of the component matrices of the triple 'realizing' the spectral factor). What is novel in the zero-free case is that the determination of the spectral factor requires no more than a *finite number of rational calculations*. These calculations involve solving a recursive discrete time matrix

Riccati equation involving $\bar{F}$, $\bar{H}^N$ and $\bar{Q}^N$ over a finite interval of length $O(n)$ or (what is nontrivially equivalent) performing a Cholesky decomposition on a (square) Toeplitz matrix with $O(nN)$ rows, formed from $\bar{F}$, $\bar{H}^N$ and $\bar{Q}^N$, see Anderson and Deistler (2009). In the event that $N$ is large and the realization of the covariance sequence is available, this is an attractive idea: with both approaches, the computational complexity is linear in $N$. Of course, it is the zero-free property of the canonical spectral factor that assures the *finite-time* convergence of the Riccati equation.

The third statement simply captures, and is also an extension of, the nesting property. Though not contained in Anderson and Deistler (2009), a proof is contained in Appendix A.

## 5. Obtaining Parameters in the AR and State-Variable Models

We suppose in this section that from the sample data $(y_t^N)$ collected over a time interval and with a large value of $N$, using PCA or an equivalent, an estimate of $(\chi_t^N)$ is obtained over effectively the same interval. This is not even a sample of a process over a finite interval, but only an estimate of it. Finding a model for the 'true' process $(\chi_t^N)$ based on the estimates then has to proceed using the assumptions that $n, q, r$ are known, that $N \geq N_0$, and that closeness of the sample-derived lagged covariance data and population lagged covariances assure closeness of the associated canonical spectral factors. Thus, the Assumptions introduced earlier need to all be in force.

Consider first the determination of the coefficient matrices in an AR model for $(\chi_t^N)$. What we say will be very little modified if we choose instead to model $f_t$, and so we omit any detailed discussion of this alternative in this section.

The fact that the spectrum of interest is generated by a (singular) autoregression as in (10) suggests that a stable miniphase spectral factor may be obtainable by Yule-Walker equations–the usual tool for obtaining an AR model (when one is known to exist) from covariance data. Indeed, this possibility was examined in Deistler et al. (2010). The Yule-Walker equations in general take the form

$$[A_1^N \ A_2^N \ldots A_N^N]\Gamma_N = [\gamma_1 \ \gamma_2 \ldots \gamma_N] \tag{11}$$

where $\gamma_j$ is the lagged covariance $\mathbb{E}[\chi_t^N(\chi_{t-j}^N)^\top]$ and $\Gamma_N$ is the $N \times N$ block Toeplitz matrix with first row $[\gamma_0 \ \gamma_1 \ldots \gamma_{N-1}]$. However, a difficulty immediately arises. For a singular covariance, there is no guarantee that $\Gamma_N$ is nonsingular. Nevertheless, it is established in Deistler et al. (2010) that the choice

$$[A_1^N \ A_2^N \ldots A_N^N] = [\gamma_1 \ \gamma_2 \ldots \gamma_N]\Gamma_N^\dagger \tag{12}$$

where use of a pseudoinverse to replace the regular inverse ensures that the resulting AR model defines a stationary process, i.e., the associated matrix polynomial $[I + A_1^N z + A_2^N z^2 + \ldots A_N^N z^N]$ has all determinantal zeros in $|z| > 1$, which is a rephrasing of the stability condition. The paper Chen et al. (2011), which imposes fewer assumptions than are incorporated here, takes this further: under some circumstances, there may be unit circle determinantal zeros, corresponding to a deterministic component of the process. (The assumptions in this paper rule out this possibility.) Further, if one does *not* use a pseudoinverse (which delivers a minimum norm solution of (11)) but uses some other procedure to define the matrices $A_i^N$ to satisfy (11), the AR system so defined may not be stable. Note that the quantities $\gamma_i$ will actually be replaced by estimates which are consistent as $N, T \to \infty$.

For completeness, we recall the potential numerical hazards arising when the key Toeplitz matrix is ill-conditioned, as discussed in Section 2.1.

The procedure for determining a state-space model is of course different. The starting point is the sample calculations giving estimates of $\mathbb{E}[\chi_t^N(\chi_{t-s}^N)^\top]$ for $s = 0, 1, 2, \ldots$. If these were exact population second moments corresponding to a system satisfying the assumptions, particularly the rational spectrum assumption, an algorithm due to Ho and

Kalman (1966) will determine a minimal dimension state variable triple realizing the sequence, i.e., a triple $\{\bar{F}, \bar{Q}^N, \bar{H}^N\}$ for which

$$\bar{H}^N \bar{F}^s \bar{Q}^N = \mathbb{E}[\chi_t^N (\chi_{t-s}^N)^\top], \quad s = 0, 1, 2, \dots$$

with $\bar{F}$ of minimal dimension. (Such a triple is only unique up to a nonsingular transformation $T$, replacing $\{\bar{F}, \bar{Q}^N, \bar{H}^N\}$ by $\{T\bar{F}T^{-1}, T\bar{Q}^N, \bar{H}^N T^{-1}\}$). Subsequently, Zeiger and McEwen (1974) presented a variant that generates approximate finite dimensional realizations of low dimension from approximate data, as needed here. The algorithm uses truncated singular value decomposition, which means that for $O(N)$-dimensioned matrices of much lower rank than $N$ the complexity appears to be $O(N)$. Of course, once the triple $\{\bar{F}, \bar{Q}^N, \bar{H}^N\}$ has been determined, the construction of $G$ using the ideas of Theorem 3 can be attempted.

Once again, it may well make more sense to seek a state variable triple realizing the sequence $\mathbb{E}[f_t (f_{t-s})^\top], \quad s = 0, 1, 2, \dots$, rather than $E[\chi_t^N (\chi_{t-s}^N)^\top], \quad s = 0, 1, 2, \dots$

The bulk of the suggested algorithms in this subsection have not been simulated with real data. However, even if they do not perform particularly well, they offer the opportunity to obtain a first estimate of a model, using which an iterative algorithm to refine the estimate can be used, Shumway and Stoffer (1982). This algorithm, based on expectation maximization, is a maximum likelihood algorithm which at each step always increases the likelihood, and necessarily therefore reaches a maximum, though it may not be a global maximum. The use of a well-chosen initialization can however promote that desired outcome.

The algorithm of Shumway and Stoffer (1982) appears to offer no straightforward technique for incorporating a rank constraint on the input coupling matrix $G$ of the canonical spectral factor, or simply the zero-free property. Consequently, while it uses Kalman filtered and smoothed estimates, it is not immediately clear how one should exploit the notion that such estimates in the zero-free case can be exactly determined using a finite number of measurements as set out in the commentary immediately following Theorem 3.

## 6. Mixed Frequency Systems

Some econometric time series may be collected monthly, others may be collected quarterly. A methodology is needed to allow graceful combination of the two periodicities, and the modelling of high dimension time series. Starting with this section, there appears a summary of ideas developed by authors Anderson and Deistler with their colleagues in treating such problems. Note importantly that mixed frequency problems are of interest outside the 'high dimensionality' context, and some limited conclusions are drawn here that do not appeal to properties such as zero freeness.

The historical roots of the applicable techniques lie in signal processing, which has treated with a technique termed 'blocking' or 'lifting' problems involving time series with integrally related periodicities, with a view to bringing to bear tools applicable to single frequency systems, such as rational transfer functions, and their descriptions and properties involving for example state-variable realizations, poles and zeros, see, e.g., (Mitra 2011, see Chapters 13 and 14). The book Bittanti and Colaneri (2009) also contains, or contains references to, many of the relevant ideas, and has a more control systems flavour.

### 6.1. Blocked Linear Systems

We draw on Chen et al. (2012) for initial ideas, which cover change of periodicity as opposed to treatment of simultaneous multiple periodicities. Once change of periodicity is understood, it is straightforward to introduce multiple periodicities.

Consider a time-invariant finite-dimensional linear system with input sequence $u_t$ and output sequence $y_t$ given by $x_{t+1} = Ax_t + Bu_t, y_t = Cx_t + Du_t$ with $t = 0, 1, 2, \dots$. Imagine then that blocks of $J$ successive inputs and outputs are collected into $N$-vectors

$U_t = [u_t^\top \ u_{t-1}^\top, \dots, u_{t-J+1}^\top]^\top$ and $Y_t = [y_t^\top \ y_{t-1}^\top, \dots, y_{t-J+1}^\top]^\top$. Then one can write the following equations for a 'blocked' or 'lifted' system:

$$x_{t+1} = A_b x_{t-J+1} + B_b U_t \tag{13}$$
$$Y_t = C_b x_{t-J+1} + D_b U_t$$

where

$$A_b = A^J \quad B_b = [B \ AB \dots A^{J-1}B] \tag{14}$$

$$C_b = \begin{bmatrix} CA^{J-1} \\ CA^{J-2} \\ \vdots \\ CA \\ C \end{bmatrix} \quad D_b = \begin{bmatrix} D & CB & \dots & C^{J-2}B \\ 0 & D & \dots & CA^{J-3}B \\ \vdots & \vdots & \ddots & \vdots \\ 0 & 0 & \dots & CB \\ 0 & 0 & \dots & D \end{bmatrix}$$

These are equations of a time-invariant linear system, of different input and output dimension but same state dimension as the original. One can regard this second system as updating every $J$ time instants, thus at (for example) $t = 0, J, 2J, \dots$. Note that even if the 'direct feedthrough' term $Du_t$ is absent in the unblocked system as when $D = 0$, a nonzero direct feedthrough term does appear in the blocked system in general. The transfer function of the blocked system is $D_b + C_b z(I - zA_b)^{-1} B_b$.

The paper Chen et al. (2011) records many connections between the original and 'blocked' transfer functions and state-variable realizations (some being novel to the paper), in particular, the carrying over of controllability and observability and minimality, connections between the normal ranks, a formula connecting the first column of the blocked transfer function and the original transfer function, relations between the poles and, separately, the finite and infinite zeros of the two transfer functions, implying that one is stable (or miniphase) if and only if the other is stable (or miniphase), and one is zero free if and only if the other is zero free.

In this connection, note that if the dimension of $y_t$ exceeds that of $u_t$, the same is true of the dimensions of $Y_t$ and $U_t$. Moreover, if the original $A, B, C, D$ matrices are generic, the unblocked and therefore the blocked system are zero free.

The property of zero-free carryover becomes relevant also in considering time series with multiple periodicities.

### 6.2. Systems with Multiple Periodicities in Their Outputs

Again our starting point is the linear system defined by $x_{t+1} = Ax_t + Bu_t, y_t = Cx_t + Du_t$, with $t = 0, 1, 2, \dots$. However we now suppose that the vector $y_t$ is partitioned into two subvectors of dimensions $p_1$ and $p_2$, thus

$$y_t = \begin{bmatrix} y_t^f \\ y_t^s \end{bmatrix}$$

The process $(y_t^f)$ is observed at all time instants, while the process $(y_t^s)$ is observed at time instants $0, J, 2J, \dots$ even though it is known to exist for all time instants. From the observer's point of view, $(y_t^f)$ and $(y_t^s)$ constitute 'fast' and 'slow' processes. Evidently the observed output comprises two (in general multivariate) time series with different integrally related periodicities. As set out in, e.g., Anderson et al. (2016a, 2016b), We can establish a single frequency system by blocking the fast process $(y_t^f)$.

In more detail, consistent with the partition of $y_t$, we also adopt the corresponding partitions

$$C = \begin{bmatrix} C^f \\ C^s \end{bmatrix} \quad D = \begin{bmatrix} D^f \\ D^s \end{bmatrix}$$

To obtain a blocked model, we block $u_t$ as before, but only use the observed values within each $y_t$ for blocking, which means that for $t = 0, J, 2J, \ldots,$

$$
U_t = \begin{bmatrix} u_t \\ u_{t-1} \\ \vdots \\ u_{t-J+1} \end{bmatrix} \qquad
Y_t = \begin{bmatrix} y_t^f \\ y_{t-1}^f \\ \vdots \\ y_{t-J+1}^f \\ y_t^s \end{bmatrix}
\tag{15}
$$

It is straightforward to obtain matrices $A_b, C_b, C_b, D_b$ constituting a state-variable realization for the system taking the sequence $U_0, U_J, \ldots$ to $Y_0, Y_J, \ldots$, see Zamani et al. (2011). In particular,

$$
A_b = A^J \quad B_b = [B \ AB \ \ldots \ A^{J-1}B]
\tag{16}
$$

$$
C_b = \begin{bmatrix} C^f A^{J-1} \\ C^f A^{J-2} \\ \vdots \\ C^f A \\ C^f \\ C^s \end{bmatrix}
\qquad
D_b = \begin{bmatrix}
D^f & C^f B & \ldots & C^f A^{J-2} B \\
0 & D^f & \ldots & C^f A^{J-3} B \\
\vdots & \vdots & \ddots & \vdots \\
0 & 0 & \ldots & C^f B \\
0 & 0 & \ldots & D^f \\
D^s & C^s B & \ldots & C^s A^{J-2} B
\end{bmatrix}
$$

$$
x_{t+1} = A_b x_{t-J+1} + B_b U_t
$$
$$
Y_t = C_b x_{t-J+1} + D_b U_t
$$

This system is self-evidently a purely time-invariant system, governed by a single period which happens to be that defined by the observed slow process $(y_t^s)$, despite the system containing within its outputs all the information from the fast process $(y_t^f)$. Of course, there is no attempt in this model to interpolate the missing (nonobserved) values of the slow process.

The original system and the blocked system have the same state-space dimension. It is natural then to ask if minimality carries over. This question is examined in Anderson et al. (2016b). If the original system is controllable, then it is straightforward to establish that there can be no nonzero vector $w$ and constant $\lambda$ for which $w^\top A_b = \lambda w^\top$, $w^\top B_b = 0$, i.e., the blocked model is controllable. The argument does not work for observability because there may be modes solely observed by $C^s$ which at the high frequency are distinct, but when sampled every $J$ time instants, merge. This can occur if for example $A$ has two distinct eigenvalues with the same $J$-th power. Such a situation is nongeneric however, and one can show that if $A$ is generic (and in particular has distinct eigenvalues whose $J$-th powers are also distinct), observability of the underlying system implies observability of the blocked system. Of course, if the underlying system is observable from $y^f$ alone, the blocked system is observable irrespective of the genericity of $A$, this following by essentially the same argument as used to study the carrying over of controllability. Furthermore, if the original system is uncontrollable, or unobservable, the same property necessarily holds for the blocked system. Evidently then, for generic $A$, the original system and the blocked system are either both minimal, or both nonminimal.

Going on from this, it follows that if the underlying system has $\{A, B, C, D\}$ and $\{TAT^{-1}, TB, CT^{-1}, D\}$ as two minimal state variable quadruples for some nonsingular $T$, the corresponding two blocked systems are $\{A_b, B_b, C_b, D_b\}$ and $\{TA_bT^{-1}, TB_b, C_bT^{-1}, D_b\}$, and if any minimal realization of the blocked system is known, it must be expressible in terms of an associated underlying system via the formula (16).

Summing up, we can state:

**Theorem 5.** *Consider an underlying linear system defined by $x_{t+1} = Ax_t + Bu_t$, $y_t = Cx_t + Du_t$, with $t = 0, 1, 2, \ldots$ and where the vector $y_t$ and matrices $C, D$ are partitioned into blocks with $p_1$ and $p_2$ rows, as*

$$y_t = \begin{bmatrix} y_t^f \\ y_t^s \end{bmatrix} \quad C = \begin{bmatrix} C^f \\ C^s \end{bmatrix} \quad D = \begin{bmatrix} D^f \\ D^s \end{bmatrix}$$

*The process $(y_t^f)$ is observed at all time instants, while the process $(y_t^s)$ is observed at time instants $0, J, 2J, \ldots$. Then the associated observed block model, with input and output given by (15), has a state-variable realization given by (16). Moreover, controllability of either the underlying system or the blocked observed system implies the same property for the other. If $A$ is generic, the same holds true in respect of observability. Coordinate basis change and blocking commute.*

We also note by way of a side remark that there is a new sort of prediction problem indirectly presenting itself here: nowcasting, which can be described roughly as prediction of the immediate but unobserved past, present or immediate future. This will apply for those time instants between those at which the slow process is measured.

Following on from the preceding theorem, it is also relevant to consider what happens when the underlying system is a canonical spectral factor generating the process $(y_t)$. Under some circumstances, the blocked system will not have this property simply because of its input, state and output dimensions, and there is more than one reason why this can happen:

1. Depending on the value of $J$ and the dimensions of $u_t, y_t$, the dimension of $U_t$ may end up larger than that of $Y_t$, i.e., the transfer function of the blocked system may be fat, as noted in Anderson et al. (2016b).
2. Depending on the value of $J$ and the dimensions of $u_t, x_t$, the dimension of $U_t$ may end up greater than that of $x_t$ (which is the state dimension of the blocked system).

Apart from this, the helpful property of zero freeness, which in many applications will be a property of the underlying system, is not guaranteed to carry over to the blocked system, although , as shown in Zamani et al. (2011), if the dimension of $y_t^f$ alone exceeds that of $u_t$, and the matrices $A, B, C, D$ are generic, then the system defined by $A_b, C_b, C_b, D_b$ is zero free.

The overarching problem is one of identifying the underlying system using the mixed frequency observations. The key is to work via the blocked system. Unsurprisingly, if the blocked system is indeed zero free, the process is much more straightforward.

In econometric applications, the condition for the zero-free property of the blocked system may well be fulfilled. However, this may not be the case, even though the underlying system is zero free. This is the main scenario treated in the next section.

## 7. Mixed Frequency System Identification

There are two broad thrusts to this section. First, we postulate that the underlying system has an AR description, and exploit that fact. Second, we exploit the structure of the blocked system (but at a certain point, also appeal to the zero-free structure of the underlying system).

### 7.1. Mixed Frequency System Identificaiton with an AR Underlying System

It is well known that the AR property is not preserved under marginalization. Therefore, even if the underlying process is AR but some of its entries are only observed every $J$ time instants, one cannot expect that the observed process is AR. (This is roughly equivalent to observing that even if the underlying process is zero-free, the blocked process may not have this property). More to the point, even if it is AR, the associated parameters in the defining matrices of the AR equation cannot be straightforwardly related to those of the underlying process. Nevertheless, using an approach based on exploiting Yule-Walker equations, progress can be made on identification. This idea was originally observed in Chen and Zadrozny (1998) and further developed in Anderson et al. (2012a). Subsequently,

it was shown in Anderson et al. (2016a) that for systems with generic parameter values, the suggested algorithm could always be executed. The key here was to ensure utilization of all observed second order moments, in contrast to Chen and Zadrozny (1998) which omitted use of some.

Suppose that the underlying process is

$$
y^t = \begin{bmatrix} y_t^f \\ y_t^s \end{bmatrix} = \begin{bmatrix} a_{ff}(1) & a_{fs}(1) \\ a_{sf}(1) & a_{ss}(1) \end{bmatrix} \begin{bmatrix} y_{t-1}^f \\ y_{t-1}^s \end{bmatrix} + \cdots +
$$
$$
+ \begin{bmatrix} a_{ff}(p) & a_{fs}(p) \\ a_{sf}(p) & a_{ss}(p) \end{bmatrix} \begin{bmatrix} y_{t-p}^f \\ y_{t-p}^s \end{bmatrix} + \begin{bmatrix} w_t^f \\ w_t^s \end{bmatrix} \tag{17}
$$

with innovations covariance

$$
\mathbb{E}\left[ \begin{bmatrix} w_t^f \\ w_t^s \end{bmatrix} [(w_t^f)^\top \ (w_t^s)^\top] \right] = \Sigma = \begin{bmatrix} \Sigma_{ff} & \Sigma_{fs} \\ \Sigma_{sf} & \Sigma_{ss} \end{bmatrix}
$$

The innovations covariance may be singular. The identifiability question is whether the coefficient matrices $a_{ff}(i), a_{fs}(i)$, etc and the entries of $\Sigma$ can be identified from covariance data associated with the time series $(y_t^f), t = 0, 1, 2, \ldots$ and $(y_t^s), t = 0, J, 2J, \ldots$.

As asserted above, this problem can be tackled by generating 'extended' Yule-Walker equations, which are obtained by multiplying (17) on the right by the fast components of $y_{t-k}^\top, k = 1, 2, \ldots$ and forming expectations. The resulting equations are linear in the AR coefficients and, crucially, it can be proved that they are generically solvable, i.e., for almost all values of the coefficient and entries of $\Sigma$, a unique solution exists, see Anderson et al. (2016a). (An example is given below). Once these coefficients have been found, the entries of $\Sigma$ can be computed, except again on a set of parameter values of measure zero. The corresponding estimators are however not "good" and in particular not asymptotically efficient, also because observed lags of the covariances of the slow process are not exploited.

*7.2. A Simple AR(1) Example*

In this subsection, and following Anderson et al. (2016b), we consider what is almost the simplest possible case of a mixed frequency AR system. The underlying system is AR(1), and has two scalar outputs, one of which is observed at every second time instant. We also assume regularity of the underlying system, so the zero-lag covariance $\Sigma$ of the driving white noise is nonsingular. We investigate to what extent the parameters of the process before observation loss can be recovered from the observed process statistics.

Before loss of observations, the process is

$$
\begin{bmatrix} y_t^f \\ y_t^s \end{bmatrix} = \begin{bmatrix} a_{ff} & a_{fs} \\ a_{sf} & a_{ss} \end{bmatrix} \begin{bmatrix} y_{t-1}^f \\ y_{t-1}^s \end{bmatrix} + \begin{bmatrix} w_t^f \\ w_t^s \end{bmatrix} \quad \mathbb{E}\left[ \begin{bmatrix} w_t^f \\ w_t^s \end{bmatrix} [w_t^f \ w_t^s] \right] = \Sigma = \begin{bmatrix} \sigma_{ff} & \sigma_{fs} \\ \sigma_{sf} & \sigma_{ss} \end{bmatrix} \tag{18}
$$

Set

$$
\mathcal{A} = \begin{bmatrix} a_{ff} & a_{fs} \\ a_{sf} & a_{ss} \end{bmatrix}
$$

With every second value of the 'slow' process $(y_t^s)$ being observed, it is easy to see that

$$
\begin{bmatrix} y_t^f \\ y_t^s \\ y_{t-1}^f \end{bmatrix} = \begin{bmatrix} \mathcal{A}^2 & 0 \\ a_{ff} a_{fs} & 0 \end{bmatrix} \begin{bmatrix} y_{t-2}^f \\ y_{t-2}^s \\ y_{t-3}^f \end{bmatrix} + \begin{bmatrix} \mathcal{A} w_{t-1} + w_t \\ w_{t-1}^f \end{bmatrix} = \tilde{\mathcal{A}} \begin{bmatrix} y_{t-2}^f \\ y_{t-2}^s \\ y_{t-3}^f \end{bmatrix} + \tilde{w}_t \tag{19}
$$

with obvious definitions of $\tilde{\mathcal{A}}$ and $\tilde{w}_t$. It is straightforward to check that

$$\tilde{\mathcal{A}} = \begin{bmatrix} a_{ff}^2 + a_{fs}a_{sf} & a_{ff}a_{fs} + a_{fs}a_{ss} & 0 \\ a_{ss}a_{ff} + a_{ss}a_{sf} & a_{sf}a_{fs} + a_{ss}^2 & 0 \\ a_{ff} & a_{fs} & 0 \end{bmatrix} \tag{20}$$

and

$$\mathbb{E}[\tilde{w}\tilde{w}^{\top}] = \tilde{\Sigma} = \begin{bmatrix} \sigma_{ff} & \sigma_{fs} & 0 \\ \sigma_{sf} & \sigma_{ss} & 0 \\ 0 & 0 & 0 \end{bmatrix} + \begin{bmatrix} a_{ff} & a_{fs} \\ a_{sf} & a_{ss} \\ 1 & 0 \end{bmatrix} \begin{bmatrix} \sigma_{ff} & \sigma_{fs} \\ \sigma_{sf} & \sigma_{ss} \end{bmatrix} \begin{bmatrix} a_{ff} & a_{sf} & 1 \\ a_{fs} & a_{ss} & 0 \end{bmatrix} \tag{21}$$

The crucial question at this point is now: given the AR process generated in (19), and an identification using the process measurements of the entries of $\tilde{\mathcal{A}}$ and $\tilde{\Sigma}$ (presumably using Yule-Walker equations which will yield unique values in the regular case), can the entries of $\mathcal{A}$ and $\Sigma$ then be inferred?

The high level answer is that for generic values of the parameters, they can indeed be inferred. The more detailed answer is that the values cannot be inferred if and only if all three of the following conditions (defining a semi-algebraic set of parameters of measure zero) are fulfilled (for details, see Anderson et al. (2016b), which builds on a simpler version of the problem assuming diagonal $\Sigma$ treated in Anderson et al. (2012a)):

$$a_{fs} = 0 \quad a_{sf} + \frac{\sigma_{sf}}{\sigma_{ff}}(a_{ss} - a_{ff}) = 0 \quad a_{ss} \neq 0$$

As noted in Anderson et al. (2016b), if the underlying process continues to have a lag of 1, the approach is generalizable to multivariate $y_t^f$ and $y_t^s$, except that the definition of nongeneric values of parameters becomes more involved.

Generalization to lags greater than 1 is however not possible using the framework given in this subsection; this is because the AR structure of the original system no longer yields an AR structure for the blocked system. This is unsurprising, given the nonclosure of AR systems under marginalization. Nevertheless, the general approach based on modified Yule-Walker equations remains available Anderson et al. (2016b).

*7.3. Mixed Frequency System Identification Exploiting Blocked System Structure*

In this subsection, we record ideas originally presented in Anderson et al. (2016b) for the case where the slow process comprises every second value in the time sequence of the relevant entries of the underlying process. However, here we describe the situation when every $J$-th value rather than every second value is used. While the underlying system is zero free, the blocked system is not assumed to have this property.

To begin, suppose the underlying system state variable realization $\{A, B, C, D\}$ is minimal. Then generically, the same is true of the blocked system, defined by the quadruple $\{A_b, B_b, C_b, D_b\}$. The blocked system itself may not be a canonical miniphase system, but as argued in Anderson et al. (2016b), there is a minimal state-variable realization of the canonical miniphase system (yielding the same spectrum) with the same $C_b$ and $A_b$. It follows that if we identify a minimal state-variable realization of the canonical miniphase system for the blocked system, and this includes state-update and state-to-output coupling

matrices $\tilde{A}_b, \tilde{C}_b$, then for some nonsingular $T$, there will hold $\tilde{A}_b = T^{-1}A_bT, \tilde{C}_b = TC_b$. From these two matrices, one seeks matrices $\tilde{A}, \tilde{C}^f, \tilde{C}^s$ such that

$$
\tilde{A}_b = \tilde{A}^J, \quad \tilde{C}_b = \begin{bmatrix} \tilde{C}^f \tilde{A}^{J-1} \\ \tilde{C}^f \tilde{A}^{J-2} \\ \vdots \\ \tilde{C}^f \tilde{A} \\ \tilde{C}^f \\ \tilde{C}^s \end{bmatrix}
\tag{22}
$$

Of course, $\tilde{C}^f$ and $\tilde{C}^s$ are immediately obtained from the second equation. With $A$ sufficiently generic that no two eigenvalues have the same $J$-th power, the matrix $\tilde{A}$ (which has the same property) is determined up to a finite number of possibilities corresponding to different possible choices for the $J$-th root of each eigenvalue of $\tilde{A}_b$. It may be that knowledge of $\tilde{C}^f \tilde{A}, \tilde{C}^f \tilde{A}^2, \ldots$ then suffices to achieve disambiguation. Alternatively the set can be disambiguated to identify just one of the possible choices by invoking a further appeal to genericity (still excluding a semi-algebraic set of measure zero, but one which contains the set ensuring that no two eigenvalues of $A$ have the same $J$-th power). Exclusion from the larger semi-algebraic set guarantees the observability of $(A, C^f)$ or equivalently of $(\tilde{A}, \tilde{C}^f)$, which is enough to secure disambiguation.

As noted above, we must be able to identify matrices $\tilde{A}_b, \tilde{C}_b$ which are part of a quadruple of matrices defining a state-variable realization of the canonical spectral factor of the blocked system. We do not however need to identify all matrices of a realization of the canonical spectral factor. We can in fact identify the relevant matrices by identifying a state-variable realization of the covariance sequence associated with the spectrum of $Y_t$—see the discussion following Theorem 3—and this is a relatively straightforward matter, using the algorithm earlier cited, Zeiger and McEwen (1974).

Now suppose in fact that the underlying system is a VAR process of order $p$, thus

$$
y_t = A_1 y_{t-1} + \cdots + A_p y_{t-p} + \nu_t
\tag{23}
$$

with $(\nu_t)$ a white noise process. Of course, the system is assumed to be stable. This system can also be described by the state-variable equations

$$
\underbrace{\begin{bmatrix} y_t \\ \vdots \\ y_{t-p+1} \end{bmatrix}}_{x_{t+1}} = \underbrace{\begin{bmatrix} A_1 & \cdots & A_{p-1} & A_p \\ I & \cdots & 0 & 0 \\ \vdots & \ddots & \vdots & \vdots \\ 0 & \cdots & I & 0 \end{bmatrix}}_{A} \underbrace{\begin{bmatrix} y_{t-1} \\ \vdots \\ y_{t-p} \end{bmatrix}}_{x_t} + \underbrace{\begin{bmatrix} b \\ 0 \\ \vdots \\ 0 \end{bmatrix}}_{B} \epsilon_t
\tag{24}
$$
$$
y_t = \underbrace{[A_1 \ldots A_p]}_{C} x_t + \underbrace{b}_{D} \epsilon_t
$$

It is explained in Anderson et al. (2016a) how the coefficient matrices $A_i$ can be determined from the observable pair $(\tilde{A}, \tilde{C})$ (where $\tilde{C} = [(\tilde{C}^f)^\top \ (\tilde{C}^s)^\top]^\top$) under the additional genericity assumptions that $A_p$ is nonsingular, as is the covariance of the white noise process $\nu_t$.

In the earlier part of this paper, we stressed the potential numerical hazards when working with AR descriptions of high dimensional processes. The same warning of course applies here if the processes are typical econometric processes. This means that procedures for reducing the problem dimensionality by exploiting the singularity of the zero lag covariance of the output are desirable. How best to do this when measured signals have two periodicities (whether or not at the blocked system level) has not however been addressed in any depth in the literature. Note though that the minimal static factors can be

constructed as easily for the mixed frequency case as for the single frequency case, since they are effectively determined using the zero lag covariance of the measured outputs. Another comparatively untreated issue for mixed frequency systems is that of passing to the common components process from a measurement process in which idiosyncratic components are additively embedded.

### 7.4. VARMA and VMA System Identification

The content of this subsection is not especially relevant to Dynamic Factor Models, but is related to the earlier material of this and the previous section, in that it focusses on further questions of identifiability when not all output values of an underlying process are observed.

An alternative starting point for the underlying process is to assume a different model class, where one postulates a model

$$y_t + A_1 y_{t-1} + \cdots + A_p y_{t-p} = B_0 w_t + B_1 w_{t-1} + \cdots + B_s w_{t-s} \tag{25}$$

with $A_i, B_i$ of dimension $N \times N$. For convenience, make the definitions

$$a(z) = I + A_1 z + \cdots + A_p z^p \quad b(z) = B_0 + B_1 z + \ldots B_s z^s \tag{26}$$

It is assumed that $y_t = [(y_t^f)^\top \ (y_t^s)^\top]^\top$ with $y_t^s$ observed at time $t = 0, J, 2J, \ldots$ for some integer $J > 1$. The stability and miniphase properties are captured by the assumptions

$$\det a(z) \neq 0, \quad \det b(z) \neq 0 \quad |z| \leq 1 \tag{27}$$

With the degrees of $a(z)$ and $b(z)$ assumed known, the identification task is to estimate the entries of the coefficient matrices in $a(z), b(z)$.

Of course, $(w_t)$ is a zero mean white noise sequence; we also assume $\mathbb{E}[w_t w_t^\top] = \Sigma$. If rank $\Sigma = q < N$, the system is singular. Otherwise it is regular.

Note that this problem setting does not have to arise in a context of high-dimensional time series (though it may), which means that non-AR modelling becomes relevant.

Identifiability of such models is linked to the question of uniqueness of the polynomials making up the transfer function $a^{-1}(z)b(z)$. In the regular case, one can require left coprimeness of the pair $a(z), b(z)$ and then enforce through multiplication of each of $a, b$ by the same unimodular (constant determinant) multiplier a canonical form. In the singular case, the situation is more complicated. For example, even for an AR system, the pair $(a, b)$ with $b(z) = B_0$ may not be left coprime. Even when they are left coprime, the Yule-Walker equations which are usually used to obtain the $A_i$ from covariances may not have a unique solution, in contrast to the regular case.

In case $s \leq p$, the reference Anderson et al. (2016a) establishes that for generic values of the entries of the coefficient matrices, identifiability can be achieved. A two-step process is used. First, analogously to the procedure outlined in Section 7.1, a modified set of Yule-Walker equations can be obtained in which only known covariance data appears, and from these, the denominator matrix $a(z)$ can be obtained, given generic values of the coefficient matrices in $a(z)$ and $b(z)$. Then by using the knowledge of all the $A_i$ and some of the covariances $\mathbb{E}[y_t y_{t-s}^\top]$, it turns out that all the missing covariance data can be obtained. Following that, the matrix polynomial $b(z)$ is straightforward to obtain.

One might wonder whether the condition $s \leq p$ is important. It appears so. One can in fact show, see Deistler et al. (2017),

**Theorem 6.** *Consider the MA model obtained from (25) by setting $A_1 = A_2 = \cdots = A_p = 0$. Suppose that $\mathbb{E}[w_t w_t^\top]$ is nonsingular. With $y_t = [(y_t^f)^\top \ (y_t^s)^\top]^\top$ and $y_t^s$ observed for $t = 0, J, 2J \ldots$ for some integer $J > 1$, the parameter matrices $B_i$ are not identifiable.*

The underlying reason for this conclusion is that there are more parameters to identify than there is relevant covariance data available to identify them.

A very simple example, drawn from Anderson et al. (2016a), establishes the point. In obvious notation, consider the following system where $y_t$ has dimension 2:

$$\begin{bmatrix} y_t^f \\ y_t^s \end{bmatrix} = \begin{bmatrix} w_t^1 \\ w_t^2 \end{bmatrix} + B_1 \begin{bmatrix} w_{t-1}^1 \\ w_{t-1}^2 \end{bmatrix} \tag{28}$$

Suppose first that all covariance data is available, viz. $\gamma(0) = \mathbb{E}[y_t y_t^\top]$ and $\gamma(1) = \mathbb{E}[y_t y_{t-1}^\top]$. It is easy to check that

$$\Sigma + B_1 \Sigma B_1^\top = \gamma(0) \quad B_1 \Sigma = \gamma(1) \tag{29}$$

Identification is the task of solving these equations for the seven unknown parameters defining $B_1$ and $\Sigma$. The number of independent scalar equations is seven; though there are multiple solutions, they can be disambiguated by the miniphase condition requiring $\det b(z) \neq 0$ when $|z| \leq 1$.

Now consider the situation where every second entry of $y_t^s$ is observed. Then the 22 entry of $\gamma(1)$ is simply unavailable, being $\mathbb{E}[y_t^s y_{t-1}^s]$. A continuum of values is possible for this entry, and hence a continuum of values for the entries of $\Sigma$ and $B_1$.

It is perhaps surprising to find in the mixed frequency case that with an underlying AR process with lag 1 one has identifiability, while with an underlying MA process with lag 1 one does not have identifiability.

## 8. Conclusions

In the course of this survey of a number of results, we have highlighted several issues that are worth re-emphasising here. Furthermore, some open issues, both in the theory and applications to data are pointed out.

First, we have recorded a number of issues arising in modelling. Though there are exceptions, most problem statements with their assumptions need to guarantee that an estimated model is continuously dependent on the data from which it is obtained. Once an explicit assumption of rationality of the model transfer function has been made, continuity of the real-valued parameters for a given (and possibly estimated) set of integer-valued specification parameters with respect to the data is almost always guaranteed. Before such a rationality assumption is made however, continuity (in the $L_\infty$ sense) of a canonical spectral factor is not automatic, and depends on some additional assumption such as boundedness of the derivative of the spectrum. It is also useful to identify which particular form of model of an entity such as a canonical spectral factor is most preferable. One can think of ARMA/AR, state-variable, or even a frequency-based description involving plots of the different entries of the spectral factor against frequency. Moreover, the proposed application for a model (and the calculations using in addressing that application) may be a determinant of the best type of model. For some applications, e.g., those involving business cycles, state-variable models exhibiting pole positions in diagonal blocks of the state update matrix, or frequency domain plots, may be clearly preferable to AR models. For forecasting, AR models are likely to be preferable. Let us recall too that while some of the assumptions used in ensuring solvability of the identification (or modelling) problem are grounded in intuitively reasonable hypotheses, others are of a more instrumentalist character, thus invoked to ensure mathematical solvability rather than to reflect some underlying natural truth.

As a second theme, we have emphasised through much of the paper both the occurrence of and benefit from the zero-free property of canonical spectral factors arising in dynamic factor modelling. It is of great utility, and deserves to be accorded significant recognition for this fact. This observation is also relevant in the mixed frequency case. Another, perhaps less important, idea flowing from linear systems considerations is the

fact that state-variable realizations of a covariance sequence and a canonical spectral factor for the associated power spectrum can be assumed to have the state transition matrix and state to output coupling matrices identical. This underpins an ability to compute from a covariance sequence a canonical spectral factor using quite different calculations to those of a Yule-Walker procedure (and thereby possibly avoid numerical problems which can arise).

A third major theme is that of mixed frequency identification problems. Several key conclusions emerge here. First, it is evident that their treatment is not as mature as the treatment of single frequency problems. Second, issues of genericity become more important. Third, certain mixed frequency identification scenarios are provably nonidentifiable (as indicated in our discussion of VARMA and MA modelling).

What now of the future? Among the outstanding issues worthy of investigation, we would certainly include the following:

1. The zero-free property has been applied to macroeconomic analysis as a means to solve the fundamentalness problem. Further work is ongoing on this and, we hope, the zero-free property will increasingly become familiar to macroeconomists, see the literature cited below Theorem 4.
2. An important observation on the zero-free property in statements like Theorem 4 is that its genericity is usually obtained with respect to a parameterization in which each parameter is free to vary independently of all the others, see the comment below Theorem 2. However, in applications to macroeconomic models, structural restrictions may prevent such free variability and can therefore interfere with the genericity of zero-freeness. For an attempt to solve this issue, see Forni et al. (2020).
3. Another observation on Theorem 4 is that an AR representation may exist also for non-generic parameters, for which zerolessness does not hold but the zeros of $B(z)$ lie outside the unit circle. However, in that case the AR would not be of finite length.
4. In some cases, we have outlined algorithms which have not yet been tried on real data. It would be worthwhile to investigate them using real data.
5. A full treatment (algorithms through to testing on real data) including the decomposition into common components and idiosyncratic processes is needed for systems with multiple frequencies.

**Author Contributions:** Conceptualization, B.D.O.A., M.D. and M.L; methodology, B.D.O.A., M.D. and M.L; formal analysis, B.D.O.A., M.D. and M.L; writing—original draft preparation, B.D.O.A.; writing—review and editing, B.D.O.A., M.D. and M.L. All authors have read and agreed to the published version of the manuscript.

**Funding:** This research received no external funding.

**Institutional Review Board Statement:** Not applicable.

**Informed Consent Statement:** Not applicable.

**Data Availability Statement:** Not applicable.

**Conflicts of Interest:** The authors declare no conflict of interest.

## Abbreviations

The following abbreviations are used in this manuscript:

| | |
|---|---|
| AIC | Akaike Information Criterion |
| AR | Autoregression |
| ARMA | Autoregression Moving Average |
| MA | Moving Average |
| PCA | Principal Components Analysis |
| SVD | Singular Value Decomposition |
| VAR | Vector Autoregression |
| VARMA | Vector Autoregression Moving Average |
| VMA | Vector Moving Average |

## Appendix A. Proof of Theorem 3, Part 3

Suppose that for any $N \geq N_0$, the transfer function $H^N(I - z\bar{F})^{-1}G$ is a spectral factor of $\Phi_\chi^N(e^{j\omega})$ with $\{\bar{F}, G, \bar{H}^N\}$ a minimal state-space realization. Define $P$ to be the unique positive definite matrix satisfying

$$P - \bar{F}P\bar{F}^\top = GG^\top.$$

Note that $P$ is positive definite since $\lambda_i(\bar{F}) < 1$ and $\bar{F}, G$ is a controllable pair. Then one can verify using the identity $P - \bar{F}P\bar{F}^\top = (I - z\bar{F})P + P(I - z^{-1}F^\top) - (I - z\bar{F})P(I - z^{-1}\bar{F}^\top)$ that

$$
\begin{aligned}
&\bar{H}^N(I - z\bar{F})^{-1}GG^\top(I - z^{-1}\bar{F}^\top)^{-1}(\bar{H}^N)^\top \\
&= \bar{H}^N(I - z\bar{F})^{-1}(P - \bar{F}P\bar{F}^\top)(I - z^{-1}\bar{F}^\top)^{-1}(\bar{H}^N)^\top \\
&= \bar{H}^N(I - z\bar{F})^{-1}[(I - z\bar{F})P + P(I - z^{-1}\bar{F}^\top) \\
&\quad - (I - z\bar{F})P(I - z^{-1}\bar{F}^\top)](I - z^{-1}\bar{F}^\top)^{-1}(\bar{H}^N)^\top \\
&= \bar{H}^N(I - z\bar{F})^{-1}P(\bar{H}^N)^\top + \bar{H}^N P(I - z^{-1}\bar{F}^\top)^{-1}(\bar{H}^N)^\top) - \bar{H}^N P(\bar{H}^N)^\top
\end{aligned}
\tag{A1}
$$

The expression on the right is also the spectrum, and so it follows that

$$\bar{H}^N(I - z\bar{F})^{-1}P(\bar{H}^N)^\top = \bar{H}^N(I - z\bar{F})^{-1}\bar{Q}^N$$

By observability of $\bar{F}, \bar{H}^N$, there must hold $P(\bar{H}^N)^\top = \bar{Q}^N$, as required.

## Notes

[1] This paper underpins a lecture by the first author presented at the 5th Vienna Workshop on High-dimensional Time Series in Macroeconomics and Finance, and celebrating the 80th birthday of Manfred Deistler.

[2] State space models have larger equivalences classes and therefore more flexibility when choosing "nice" representatives. Special representatives, in echelon form, can yield state space models with the same parameters as ARMA models.

[3] This is an easy consequence of the celebrated solution of the partial realization problem for covariance sequences, Byrnes and Lindquist (1997).

[4] In a practical situation, given that sensors normally are not noise free, the introduction of more sensors is likely to aid the de-noising process. Our remarks here pertain to the process of common components.

[5] Such a spanning set of entries can be obtained by identifying the rows defining an $r \times r$ nonsingular principal submatrix of $\mathbb{E}[\chi_t^N(\chi_t^N)^\top]$

[6] As an alternative, PCA can also be used to obtain a minimal static factor.

[7] For a scalar rational transfer function written as a ratio of coprime polynomials $n(z)/d(z)$, the zero free property is equivalent to $n(z)$ being a nonzero constant. The idea generalizes to coprime matrix fraction representations of a rational transfer function matrix

[8] Coprimeness means there exists no polynomial $C^N(z) \in \mathbb{R}^{N \times N}[z]$ with nonconstant determinant and matrices $\tilde{A}^N \in \mathbb{R}^{N \times N}[z], \tilde{B}^N \in \mathbb{R}^{N \times q}[z]$ such that $\bar{A}^N = C^N \tilde{A}^N$ and $\bar{B}^N = C^N \tilde{B}^N$

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
