# Peer review of "Linear System Challenges of Dynamic Factor Models"

_econometrics, doi:10.3390/econometrics10040035_

Round 1
Reviewer 1 Report
See attachment

Reviewer 2 Report
This is a nice paper surveying the area of dynamic factor models. The survey is reasonably different from another recent survey of the same group of authors. I also do see merits in publishing such a condensed view of the developments in the field.
While I think the paper is clearly structured in general I find the introduction to be somewhat unfocused. I would put the first paragraph in a footnote as it is not directly of interest for most readers. The rest of the introduction is more a summary of the structure of the paper, but does not convey to the reader nicely why the paper should be read, what the main focus of the paper is. What need does the paper react to? Why do the authors think that surveys on this topic are in order? I would really like to read such comments and miss them in the introduction.
Also the paper is not explicit on the typical sources of data for building such models. Given that the paper is to be published in an econometrics journal the authors most likely have economic data in mind. Some of the examples in the paper also hint in that direction. From my point of view it would benefit the paper if some ideas on typical data sets could be used in the presentation of the models.
This comments also relates to the modeling goals described in section 2.1. It is hard to appreciate this section without having at least some prototypical applications in mind.
Additionally in particular for this part it is hard for the reader to follow the ideas. One can appreciate the comments, but the relation to the topic of DFM is harder to understand here. I suppose that the main function of this section is to lead to the imposition of the assumptions; to argument as to why imposing assumptions such as smoothness of the spectrum might be of interest; why the usage of models having a finite number of parameters is not a strong restriction of generality (even if letting N tend to infinity would suggest otherwise).
In this respect I would suggest to make these points more clearly visible for the readers in these sections. In particular -- as maybe not many readers of this journal will be aware of the engineering literature -- the issues related to the sensitivity of the spectral density as a function of the covariance sequence in the non-rational case (as well as the sensitivity with respect to the pole locations in the rational case) might be illustrated with examples or more explicit references to such examples to make the ideas clearer.
Other than that I only do have a number of minor comments indicated using the line numbers in the margin of the paper:
+ l. 52 ff. Maybe it would be a good idea to state here that only stationary processes are dealt with.
+ l. 89: I am not sure that the term "rational model" is appropriate here as before a more general model than involving transfer functions is mentioned.
+ l. 144: Is this function part of the signal processing toolbox or core MATLAB?
+ l. 170: In the formal differentiation I get an extra term (-j). This is immaterial for the norm bound afterwards.
+ l. 210: A dot too much.
+ l. 319: I am not sure whether the term "first order process" is common knowledge in econometrics.
+ l. 339 and also below "take generic values" or "are generic". I think the usage of this term is somewhat ambiguous. I can understand that the idea is to state that for generic values of the matrices in an appropriate space certain properties hold. However, without mentioning the properties which occur generically, the term is void. Another generic property is that all eigenvalues of F are nonzero, but this is not meant. Or that certain entries of the matrices are nonzero. I would suggest either the explicit definition of a set within which the different genericity properties hold. Or to always be explicit which property is assumed to hold generically when the term is mentioned.
+ l. 341: From my point of view it is unusual to include comments in the main part of a theorem. I would put this comment into a footnote.
+ footnote 7 on p. 12: This footnote is hard to understand as it mixes \Gamma and \Gamma_N. For finite N one does not need distinct eigenvalues for the consistency. Continuity of eigenprojections also holds for multiple singular values. For N tending to infinity the situation gets more complicated as \Gamma is typically not a compact operator and hence not approximable by the usual embedding of \Gamma_N consistently. A noncompact operator may have a spectrum more complex than just a pointspectrum.
+ l. 631: A tilde is missing on the first A.
+ l. 664: The sentence "It also reports on ..." does have no function.
+ l. 710: "their assumptions need to guarantee"
+ l. 714: "Before such a rationality assumption is made however, the observation is important." I do not understand this sentence. The observation of what?
+ Conclusions: If one reads the conclusions the main impression is that the theory is there and one only has to take it to data. Are there no open questions? One issue I can think of immediately is the question of genericity. Maybe this is not needed and one can find AR representations also in the non-generic cases. Also one might think of situations where genericity interferes with structural restrictions. Typically these restrict the model set to a lower dimensional manifold. Are there any guarantees that this does not conflict with generic properties? Are there any discussions of such problems in the literature?
+ Author Contributions: Marco Lippi is constantly abbreviated as M.C.
Round 2
Reviewer 2 Report
The revision has taken up my major points. I do not have any other reservations except for a small number of typos:
l. 110: "freedom to choose a coordinate basis."
l. 248: A dot too much.
below l. 398: under 1. There appears to be a mixup of symbols with bars and without.
l. 773: "sense)" closing bracket missing.
Author Response
The reviewer identified four typo-level matters requiring attention. The required adjustments were made to the manuscript.
No other matters were raised.